# SGDR: Stochastic Gradient Descent with Warm Restarts

**Ilya Loshchilov & Frank Hutter**
University of Freiburg
Freiburg, Germany,
{ilya,fh}@cs.uni-freiburg.de

## Abstract

Restart techniques are common in gradient-free optimization to deal with multi-modal functions. Partial warm restarts are also gaining popularity in gradient-based optimization to improve the rate of convergence in accelerated gradient schemes to deal with ill-conditioned functions. In this paper, we propose a simple warm restart technique for stochastic gradient descent to improve its anytime performance when training deep neural networks. We empirically study its performance on the CIFAR-10 and CIFAR-100 datasets, where we demonstrate new state-of-the-art results at 3.14% and 16.21%, respectively. We also demonstrate its advantages on a dataset of EEG recordings and on a downsampled version of the ImageNet dataset. Our source code is available at
https://github.com/loshchil/SGDR

## 1 Introduction

Deep neural networks (DNNs) are currently the best-performing method for many classification problems, such as object recognition from images (Krizhevsky et al., 2012a; Donahue et al., 2014) or speech recognition from audio data (Deng et al., 2013). Their training on large datasets (where DNNs perform particularly well) is the main computational bottleneck: it often requires several days, even on high-performance GPUs, and any speedups would be of substantial value.

The training of a DNN with $n$ free parameters can be formulated as the problem of minimizing a function $f : \mathbb{R}^n \to \mathbb{R}$. The commonly used procedure to optimize $f$ is to iteratively adjust $\boldsymbol{x}_t \in \mathbb{R}^n$ (the parameter vector at time step $t$) using gradient information $\nabla f_t(\boldsymbol{x}_t)$ obtained on a relatively small $t$-th batch of $b$ datapoints. The Stochastic Gradient Descent (SGD) procedure then becomes an extension of the Gradient Descent (GD) to stochastic optimization of $f$ as follows:

$$\boldsymbol{x}_{t+1} = \boldsymbol{x}_t - \eta_t \nabla f_t(\boldsymbol{x}_t), \tag{1}$$

where $\eta_t$ is a learning rate. One would like to consider second-order information

$$\boldsymbol{x}_{t+1} = \boldsymbol{x}_t - \eta_t \boldsymbol{H}_t^{-1} \nabla f_t(\boldsymbol{x}_t), \tag{2}$$

but this is often infeasible since the computation and storage of the inverse Hessian $\boldsymbol{H}_t^{-1}$ is intractable for large $n$. The usual way to deal with this problem by using limited-memory quasi-Newton methods such as L-BFGS (Liu & Nocedal, 1989) is not currently in favor in deep learning, not the least due to (i) the stochasticity of $\nabla f_t(\boldsymbol{x}_t)$, (ii) ill-conditioning of $f$ and (iii) the presence of saddle points as a result of the hierarchical geometric structure of the parameter space (Fukumizu & Amari, 2000). Despite some recent progress in understanding and addressing the latter problems (Bordes et al., 2009; Dauphin et al., 2014; Choromanska et al., 2014; Dauphin et al., 2015), state-of-the-art optimization techniques attempt to approximate the inverse Hessian in a reduced way, e.g., by considering only its diagonal to achieve adaptive learning rates. AdaDelta (Zeiler, 2012) and Adam (Kingma & Ba, 2014) are notable examples of such methods.

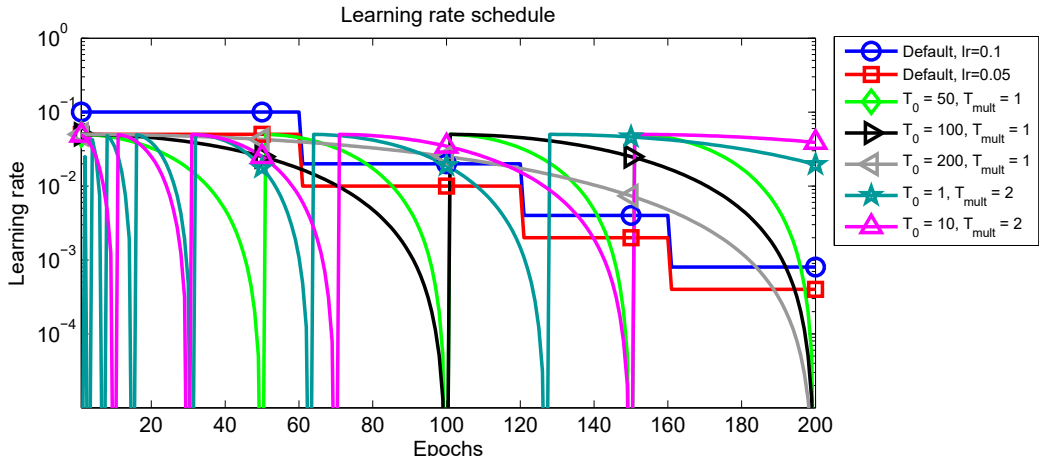

Figure 1: Alternative schedule schemes of learning rate $\eta_t$ over batch index $t$: default schemes with $\eta_0 = 0.1$ (blue line) and $\eta_0 = 0.05$ (red line) as used by Zagoruyko & Komodakis (2016); warm restarts simulated every $T_0 = 50$ (green line), $T_0 = 100$ (black line) and $T_0 = 200$ (grey line) epochs with $\eta_t$ decaying during $i$-th run from $\eta_{max}^i = 0.05$ to $\eta_{min}^i = 0$ according to eq. (5); warm restarts starting from epoch $T_0 = 1$ (dark green line) and $T_0 = 10$ (magenta line) with doubling ($T_{mult} = 2$) periods $T_i$ at every new warm restart.

Intriguingly enough, the current state-of-the-art results on CIFAR-10, CIFAR-100, SVHN, ImageNet, PASCAL VOC and MS COCO datasets were obtained by Residual Neural Networks (He et al., 2015; Huang et al., 2016c; He et al., 2016; Zagoruyko & Komodakis, 2016) trained without the use of advanced methods such as AdaDelta and Adam. Instead, they simply use SGD with momentum [1]:

$$\boldsymbol{v}_{t+1} = \mu_t \boldsymbol{v}_t - \eta_t \nabla f_t(\boldsymbol{x}_t), \tag{3}$$

$$\boldsymbol{x}_{t+1} = \boldsymbol{x}_t + \boldsymbol{v}_{t+1}, \tag{4}$$

where $\boldsymbol{v}_t$ is a velocity vector initially set to $\boldsymbol{0}$, $\eta_t$ is a decreasing learning rate and $\mu_t$ is a momentum rate which defines the trade-off between the current and past observations of $\nabla f_t(\boldsymbol{x}_t)$. The main difficulty in training a DNN is then associated with the scheduling of the learning rate and the amount of L2 weight decay regularization employed. A common learning rate schedule is to use a constant learning rate and divide it by a fixed constant in (approximately) regular intervals. The blue line in Figure 1 shows an example of such a schedule, as used by Zagoruyko & Komodakis (2016) to obtain the state-of-the-art results on CIFAR-10, CIFAR-100 and SVHN datasets.

In this paper, we propose to periodically simulate warm restarts of SGD, where in each restart the learning rate is initialized to some value and is scheduled to decrease. Four different instantiations of this new learning rate schedule are visualized in Figure 1. Our empirical results suggest that SGD with warm restarts requires $2\times$ to $4\times$ fewer epochs than the currently-used learning rate schedule schemes to achieve comparable or even better results. Furthermore, combining the networks obtained right before restarts in an ensemble following the approach proposed by Huang et al. (2016a) improves our results further to 3.14% for CIFAR-10 and 16.21% for CIFAR-100. We also demonstrate its advantages on a dataset of EEG recordings and on a downsampled version of the ImageNet dataset.

---

[1]More specifically, they employ Nesterov's momentum (Nesterov, 1983; 2013)

## 2 RELATED WORK

### 2.1 RESTARTS IN GRADIENT-FREE OPTIMIZATION

When optimizing multimodal functions one may want to find all global and local optima. The tractability of this task depends on the landscape of the function at hand and the budget of function evaluations. Gradient-free optimization approaches based on niching methods (Preuss, 2015) usually can deal with this task by covering the search space with dynamically allocated niches of local optimizers. However, these methods usually work only for relatively small search spaces, e.g., $n < 10$, and do not scale up due to the curse of dimensionality (Preuss, 2010). Instead, the current state-of-the-art gradient-free optimizers employ various restart mechanisms (Hansen, 2009; Loshchilov et al., 2012). One way to deal with multimodal functions is to iteratively sample a large number $\lambda$ of candidate solutions, make a step towards better solutions and slowly shape the sampling distribution to maximize the likelihood of successful steps to appear again (Hansen & Kern, 2004). The larger the $\lambda$, the more global search is performed requiring more function evaluations. In order to achieve good anytime performance, it is common to start with a small $\lambda$ and increase it (e.g., by doubling) after each restart. This approach works best on multimodal functions with a global funnel structure and also improves the results on ill-conditioned problems where numerical issues might lead to premature convergence when $\lambda$ is small (Hansen, 2009).

### 2.2 RESTARTS IN GRADIENT-BASED OPTIMIZATION

Gradient-based optimization algorithms such as BFGS can also perform restarts to deal with multimodal functions (Ros, 2009). In large-scale settings when the usual number of variables $n$ is on the order of $10^3 - 10^9$, the availability of gradient information provides a speedup of a factor of $n$ w.r.t. gradient-free approaches. Warm restarts are usually employed to improve the convergence rate rather than to deal with multimodality: often it is sufficient to approach any local optimum to a given precision and in many cases the problem at hand is unimodal. Fletcher & Reeves (1964) proposed to flesh the history of conjugate gradient method every $n$ or $(n + 1)$ iterations. Powell (1977) proposed to check whether enough orthogonality between $\nabla f(\boldsymbol{x}_{t-1})$ and $\nabla f(\boldsymbol{x}_t)$ has been lost to warrant another warm restart. Recently, O'Donoghue & Candes (2012) noted that the iterates of accelerated gradient schemes proposed by Nesterov (1983; 2013) exhibit a periodic behavior if momentum is overused. The period of the oscillations is proportional to the square root of the local condition number of the (smooth convex) objective function. The authors showed that fixed warm restarts of the algorithm with a period proportional to the conditional number achieves the optimal linear convergence rate of the original accelerated gradient scheme. Since the condition number is an unknown parameter and its value may vary during the search, they proposed two adaptive warm restart techniques (O'Donoghue & Candes, 2012):

- **The function scheme** restarts whenever the objective function increases.

- **The gradient scheme** restarts whenever the angle between the momentum term and the negative gradient is obtuse, i.e, when the momentum seems to be taking us in a bad direction, as measured by the negative gradient at that point. This scheme resembles the one of Powell (1977) for the conjugate gradient method.

O'Donoghue & Candes (2012) showed (and it was confirmed in a set of follow-up works) that these simple schemes provide an acceleration on smooth functions and can be adjusted to accelerate state-of-the-art methods such as FISTA on nonsmooth functions.

Smith (2015; 2016) recently introduced cyclical learning rates for deep learning, his approach is closely-related to our approach in its spirit and formulation but does not focus on restarts.

Yang & Lin (2015) showed that Stochastic subGradient Descent with restarts can achieve a *linear convergence rate* for a class of non-smooth and non-strongly convex optimization problems where the epigraph of the objective function is a polyhedron. In contrast to our work, they never increase the learning rate to perform restarts but decrease it geometrically at each epoch. To perform restarts, they periodically reset the current solution to the averaged solution from the previous epoch.

## 3 STOCHASTIC GRADIENT DESCENT WITH WARM RESTARTS (SGDR)

The existing restart techniques can also be used for stochastic gradient descent if the stochasticity is taken into account. Since gradients and loss values can vary widely from one batch of the data to another, one should denoise the incoming information: by considering averaged gradients and losses, e.g., once per epoch, the above-mentioned restart techniques can be used again.

In this work, we consider one of the simplest warm restart approaches. We simulate a new warm-started run / restart of SGD once $T_i$ epochs are performed, where $i$ is the index of the run. Importantly, the restarts are not performed from scratch but emulated by increasing the learning rate $\eta_t$ while the old value of $\boldsymbol{x}_t$ is used as an initial solution. The amount of this increase controls to which extent the previously acquired information (e.g., momentum) is used.

Within the $i$-th run, we decay the learning rate with a cosine annealing for each batch as follows:

$$\eta_t = \eta_{min}^i + \frac{1}{2}(\eta_{max}^i - \eta_{min}^i)(1 + \cos(\frac{T_{cur}}{T_i}\pi)), \tag{5}$$

where $\eta_{min}^i$ and $\eta_{max}^i$ are ranges for the learning rate, and $T_{cur}$ accounts for how many epochs have been performed since the last restart. Since $T_{cur}$ is updated at each batch iteration $t$, it can take discredited values such as 0.1, 0.2, etc. Thus, $\eta_t = \eta_{max}^i$ when $t = 0$ and $T_{cur} = 0$. Once $T_{cur} = T_i$, the cos function will output $-1$ and thus $\eta_t = \eta_{min}^i$. The decrease of the learning rate is shown in Figure 1 for fixed $T_i = 50$, $T_i = 100$ and $T_i = 200$; note that the logarithmic axis obfuscates the typical shape of the cosine function.

In order to improve anytime performance, we suggest an option to start with an initially small $T_i$ and increase it by a factor of $T_{mult}$ at every restart (see, e.g., Figure 1 for $T_0 = 1, T_{mult} = 2$ and $T_0 = 10, T_{mult} = 2$). It might be of great interest to decrease $\eta_{max}^i$ and $\eta_{min}^i$ at every new restart. However, for the sake of simplicity, here, we keep $\eta_{max}^i$ and $\eta_{min}^i$ the same for every $i$ to reduce the number of hyperparameters involved.

Since our simulated warm restarts (the increase of the learning rate) often temporarily worsen performance, we do not always use the last $\boldsymbol{x}_t$ as our recommendation for the best solution (also called the *incumbent solution*). While our recommendation during the first run (before the first restart) is indeed the last $\boldsymbol{x}_t$, our recommendation after this is *a solution obtained at the end of the last performed run at $\eta_t = \eta_{min}^i$*. We emphasize that with the help of this strategy, our method does not require a separate validation data set to determine a recommendation.

## 4 EXPERIMENTAL RESULTS

### 4.1 EXPERIMENTAL SETTINGS

We consider the problem of training Wide Residual Neural Networks (WRNs; see Zagoruyko & Komodakis (2016) for details) on the CIFAR-10 and CIFAR-100 datasets (Krizhevsky, 2009). We will use the abbreviation WRN-$d$-$k$ to denote a WRN with depth $d$ and width $k$. Zagoruyko & Komodakis (2016) obtained the best results with a WRN-28-10 architecture, i.e., a Residual Neural Network with $d = 28$ layers and $k = 10$ times more filters per layer than used in the original Residual Neural Networks (He et al., 2015; 2016).

The CIFAR-10 and CIFAR-100 datasets (Krizhevsky, 2009) consist of $32\times32$ color images drawn from 10 and 100 classes, respectively, split into 50,000 train and 10,000 test images. For image preprocessing Zagoruyko & Komodakis (2016) performed global contrast normalization and ZCA whitening. For data augmentation they performed horizontal flips and random crops from the image padded by 4 pixels on each side, filling missing pixels with reflections of the original image.

For training, Zagoruyko & Komodakis (2016) used SGD with Nesterov's momentum with initial learning rate set to $\eta_0 = 0.1$, weight decay to 0.0005, dampening to 0, momentum to 0.9 and minibatch size to 128. The learning rate is dropped by a factor of 0.2 at 60, 120 and 160 epochs, with a total budget of 200 epochs. We reproduce the results of Zagoruyko & Komodakis (2016) with the same settings except that i) we subtract per-pixel mean only and do not use ZCA whitening; ii) we use SGD with momentum as described by eq. (3-4) and not Nesterov's momentum.

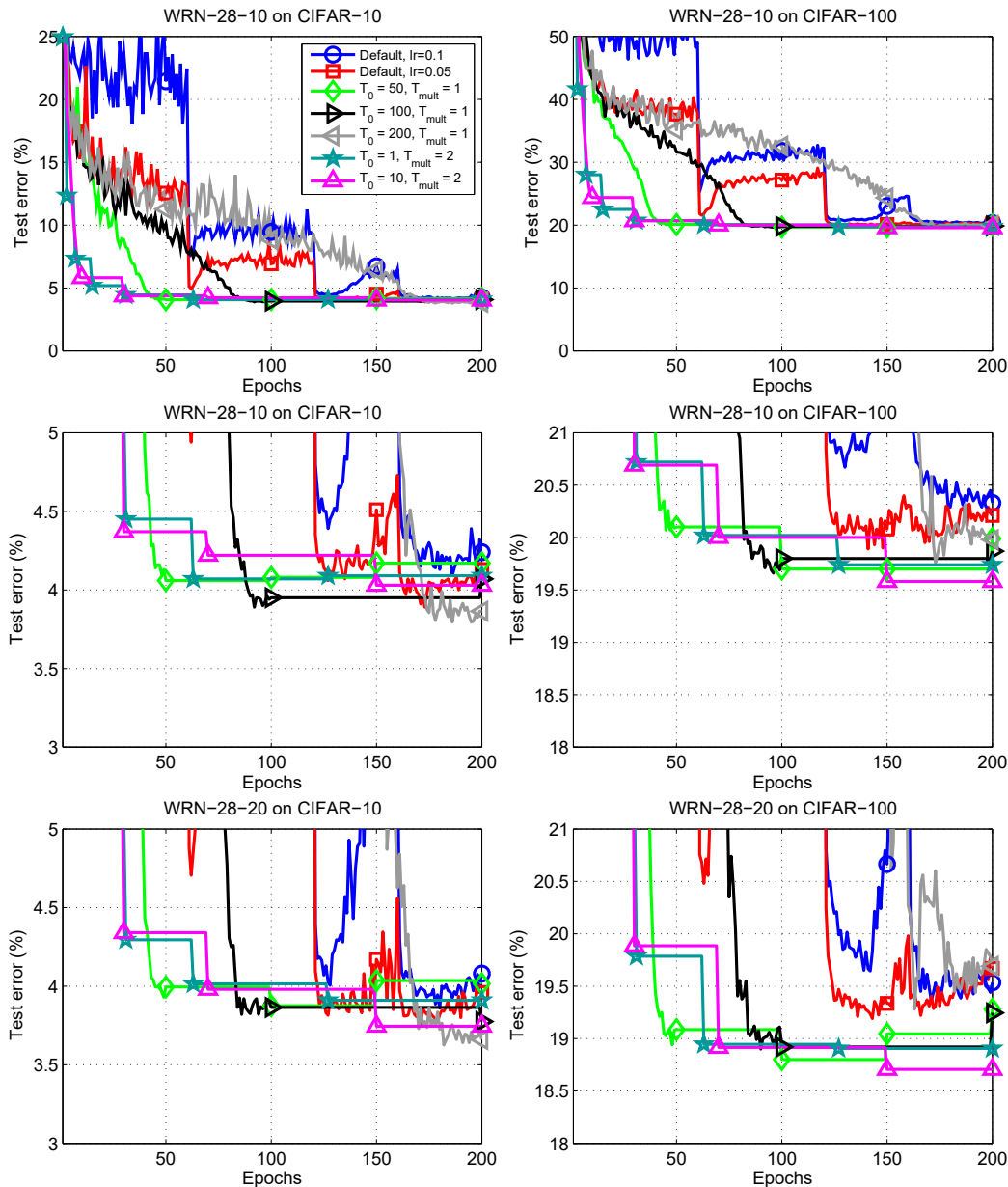

Figure 2: Test errors on CIFAR-10 (left column) and CIFAR-100 (right column) datasets. Note that for SGDR we only plot the recommended solutions. The top and middle rows show the same results on WRN-28-10, with the middle row zooming into the good performance region of low test error. The bottom row shows performance with a wider network, WRN-28-20.

The results of the default learning rate schedules of Zagoruyko & Komodakis (2016) with $\eta_0 = 0.1$ and $\eta_0 = 0.05$ are depicted by the blue and red lines, respectively. The schedules of $\eta_t$ used in SGDR are shown with i) restarts every $T_0 = 50$ epochs (green line); ii) restarts every $T_0 = 100$ epochs (black line); iii) restarts every $T_0 = 200$ epochs (gray line); iv) restarts with doubling ($T_{mult} = 2$) periods of restarts starting from the first epoch ($T_0 = 1$, dark green line); and v) restarts with doubling ($T_{mult} = 2$) periods of restarts starting from the tenth epoch ($T_0 = 10$, magenta line).

The schedule of $\eta_t$ used by Zagoruyko & Komodakis (2016) is depicted by the blue line in Figure 1. The same schedule but with $\eta_0 = 0.05$ is depicted by the red line. The schedule of $\eta_t$ used in SGDR is also shown in Figure 1, with two initial learning rates $T_0$ and two restart doubling periods.

| | depth-$k$ | # params | # runs | CIFAR-10 | CIFAR-100 |
|---|---|---|---|---|---|
| original-ResNet (He et al., 2015) | 110 | 1.7M | mean of 5 | 6.43 | 25.16 |
| | 1202 | 10.2M | mean of 5 | 7.93 | 27.82 |
| stoc-depth (Huang et al., 2016c) | 110 | 1.7M | 1 run | 5.23 | 24.58 |
| | 1202 | 10.2M | 1 run | 4.91 | n/a |
| pre-act-ResNet (He et al., 2016) | 110 | 1.7M | med. of 5 | 6.37 | n/a |
| | 164 | 1.7M | med. of 5 | 5.46 | 24.33 |
| | 1001 | 10.2M | med. of 5 | 4.62 | 22.71 |
| WRN (Zagoruyko & Komodakis, 2016) | 16-8 | 11.0M | 1 run | 4.81 | 22.07 |
| | 28-10 | 36.5M | 1 run | 4.17 | 20.50 |
| with dropout | 28-10 | 36.5M | 1 run | n/a | 20.04 |
| WRN (ours) | | | | | |
| default with $\eta_0 = 0.1$ | 28-10 | 36.5M | med. of 5 | 4.24 | 20.33 |
| default with $\eta_0 = 0.05$ | 28-10 | 36.5M | med. of 5 | 4.13 | 20.21 |
| $T_0 = 50, T_{mult} = 1$ | 28-10 | 36.5M | med. of 5 | 4.17 | 19.99 |
| $T_0 = 100, T_{mult} = 1$ | 28-10 | 36.5M | med. of 5 | 4.07 | 19.87 |
| $T_0 = 200, T_{mult} = 1$ | 28-10 | 36.5M | med. of 5 | 3.86 | 19.98 |
| $T_0 = 1, T_{mult} = 2$ | 28-10 | 36.5M | med. of 5 | 4.09 | 19.74 |
| $T_0 = 10, T_{mult} = 2$ | 28-10 | 36.5M | med. of 5 | 4.03 | 19.58 |
| default with $\eta_0 = 0.1$ | 28-20 | 145.8M | med. of 2 | 4.08 | 19.53 |
| default with $\eta_0 = 0.05$ | 28-20 | 145.8M | med. of 2 | 3.96 | 19.67 |
| $T_0 = 50, T_{mult} = 1$ | 28-20 | 145.8M | med. of 2 | 4.01 | 19.28 |
| $T_0 = 100, T_{mult} = 1$ | 28-20 | 145.8M | med. of 2 | **3.77** | 19.24 |
| $T_0 = 200, T_{mult} = 1$ | 28-20 | 145.8M | med. of 2 | **3.66** | 19.69 |
| $T_0 = 1, T_{mult} = 2$ | 28-20 | 145.8M | med. of 2 | 3.91 | **18.90** |
| $T_0 = 10, T_{mult} = 2$ | 28-20 | 145.8M | med. of 2 | **3.74** | **18.70** |

Table 1: Test errors of different methods on CIFAR-10 and CIFAR-100 with moderate data augmentation (flip/translation). In the second column $k$ is a widening factor for WRNs. Note that the computational and memory resources used to train all WRN-28-10 are the same. In all other cases they are different, but WRNs are usually faster than original ResNets to achieve the same accuracy (e.g., up to a factor of 8 according to Zagoruyko & Komodakis (2016)). Bold text is used only to highlight better results and is not based on statistical tests (too few runs).

## 4.2 SINGLE-MODEL RESULTS

Table 1 shows that our experiments reproduce the results given by Zagoruyko & Komodakis (2016) for WRN-28-10 both on CIFAR-10 and CIFAR-100. These "default" experiments with $\eta_0 = 0.1$ and $\eta_0 = 0.05$ correspond to the blue and red lines in Figure 2. The results for $\eta_0 = 0.05$ show better performance, and therefore we use $\eta_0 = 0.05$ in our later experiments.

SGDR with $T_0 = 50$, $T_0 = 100$ and $T_0 = 200$ for $T_{mult} = 1$ perform warm restarts every 50, 100 and 200 epochs, respectively. A single run of SGD with the schedule given by eq. (5) for $T_0 = 200$ shows the best results suggesting that the original schedule of WRNs might be suboptimal w.r.t. the test error in these settings. However, the same setting with $T_0 = 200$ leads to the worst anytime performance except for the very last epochs.

SGDR with $T_0 = 1, T_{mult} = 2$ and $T_0 = 10, T_{mult} = 2$ performs its first restart after 1 and 10 epochs, respectively. Then, it doubles the maximum number of epochs for every new restart. The main purpose of this doubling is to reach good test error as soon as possible, i.e., achieve good anytime performance. Figure 2 shows that this is achieved and test errors around 4% on CIFAR-10 and around 20% on CIFAR-100 can be obtained about 2-4 times faster than with the default schedule used by Zagoruyko & Komodakis (2016).

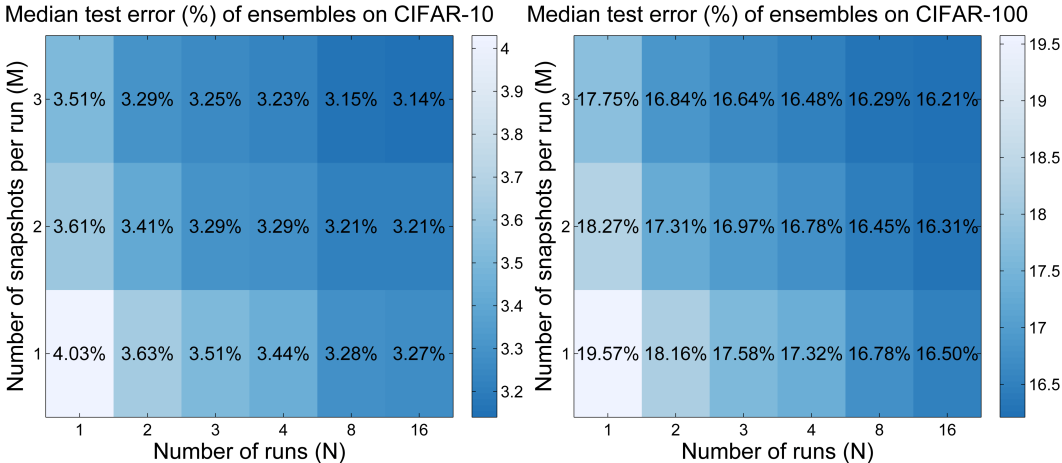

Figure 3: Test errors of ensemble models built from $N$ runs of SGDR on WRN-28-10 with $M$ model snapshots per run made at epochs 150, 70 and 30 (right before warm restarts of SGDR as suggested by Huang et al. (2016a)). When $M$=1 (respectively, $M$=2), we aggregate probabilities of softmax layers of snapshot models at epoch index 150 (respectively, at epoch indexes 150 and 70).

|  | CIFAR-10 | CIFAR-100 |
|---|---|---|
| $N = 1$ run of WRN-28-10 with $M = 1$ snapshot (median of 16 runs) | 4.03 | 19.57 |
| $N = 1$ run of WRN-28-10 with $M = 3$ snapshots per run | 3.51 | 17.75 |
| $N = 3$ runs of WRN-28-10 with $M = 3$ snapshots per run | 3.25 | 16.64 |
| $N = 16$ runs of WRN-28-10 with $M = 3$ snapshots per run | **3.14** | **16.21** |

Table 2: Test errors of ensemble models on CIFAR-10 and CIFAR-100 datasets.

Since SGDR achieves good performance faster, it may allow us to train larger networks. We therefore investigated whether results on CIFAR-10 and CIFAR-100 can be further improved by making WRNs two times wider, i.e., by training WRN-28-20 instead of WRN-28-10. Table 1 shows that the results indeed improved, by about 0.25% on CIFAR-10 and by about 0.5-1.0% on CIFAR-100. While network architecture WRN-28-20 requires roughly three-four times more computation than WRN-28-10, the aggressive learning rate reduction of SGDR nevertheless allowed us to achieve a better error rate in the same time on WRN-28-20 as we spent on 200 epochs of training on WRN-28-10. Specifically, Figure 2 (right middle and right bottom) show that after only 50 epochs, SGDR (even without restarts, using $T_0 = 50, T_{mult} = 1$) achieved an error rate below 19% (whereas none of the other learning methods performed better than 19.5% on WRN-28-10). We therefore have hope that – by enabling researchers to test new architectures faster – SGDR's good anytime performance may also lead to improvements of the state of the art.

In a final experiment for SGDR by itself, Figure 7 in the appendix compares SGDR and the default schedule with respect to training and test performance. As the figure shows, SGDR optimizes training loss faster than the standard default schedule until about epoch 120. After this, the default schedule overfits, as can be seen by an increase of the test error both on CIFAR-10 and CIFAR-100 (see, e.g., the right middle plot of Figure 7). In contrast, we only witnessed very mild overfitting for SGDR.

## 4.3 Ensemble Results

Our initial arXiv report on SGDR (Loshchilov & Hutter, 2016) inspired a follow-up study by Huang et al. (2016a) in which the authors suggest to take $M$ snapshots of the models obtained by SGDR (in their paper referred to as cyclical learning rate schedule and cosine annealing cycles) right before $M$ last restarts and to use those to build an ensemble, thereby obtaining ensembles "for free" (in contrast to having to perform multiple independent runs). The authors demonstrated new state-of-

the-art results on CIFAR datasets by making ensembles of DenseNet models (Huang et al., 2016b). Here, we investigate whether their conclusions hold for WRNs used in our study. We used WRN-28-10 trained by SGDR with $T_0 = 10, T_{mult} = 2$ as our baseline model.

Figure 3 and Table 2 aggregate the results of our study. The original test error of 4.03% on CIFAR-10 and 19.57% on CIFAR-100 (median of 16 runs) can be improved to 3.51% on CIFAR-10 and 17.75% on CIFAR-100 when $M = 3$ snapshots are taken at epochs 30, 70 and 150: when the learning rate of SGDR with $T_0 = 10, T_{mult} = 2$ is scheduled to achieve 0 (see Figure 1) and the models are used with uniform weights to build an ensemble. To achieve the same result, one would have to aggregate $N = 3$ models obtained at epoch 150 of $N = 3$ independent runs (see $N = 3, M = 1$ in Figure 3). Thus, the aggregation from snapshots provides a 3-fold speedup in these settings because additional ($M > 1$-th) snapshots from a single SGDR run are computationally free. Interestingly, aggregation of models from independent runs (when $N > 1$ and $M = 1$) does not scale up as well as from $M > 1$ snapshots of independent runs when the same number of models is considered: the case of $N = 3$ and $M = 3$ provides better performance than the cases of $M = 1$ with $N = 18$ and $N = 21$. Not only the number of snapshots $M$ per run but also their origin is crucial. Thus, naively building ensembles from models obtained at last epochs only (i.e., $M = 3$ snapshots at epochs 148, 149, 150) did not improve the results (i.e., the baseline of $M = 1$ snapshot at 150) thereby confirming the conclusion of Huang et al. (2016a) that snapshots of SGDR provide a useful diversity of predictions for ensembles.

Three runs ($N = 3$) of SGDR with $M = 3$ snapshots per run are sufficient to greatly improve the results to 3.25% on CIFAR-10 and 16.64% on CIFAR-100 outperforming the results of Huang et al. (2016a). By increasing $N$ to 16 one can achieve 3.14% on CIFAR-10 and 16.21% on CIFAR-100. We believe that these results could be further improved by considering better baseline models than WRN-28-10 (e.g., WRN-28-20).

## 4.4 EXPERIMENTS ON A DATASET OF EEG RECORDINGS

To demonstrate the generality of SGDR, we also considered a very different domain: a dataset of electroencephalographic (EEG) recordings of brain activity for classification of actual right and left hand and foot movements of 14 subjects with roughly 1000 trials per subject. The best classification results obtained with the original pipeline based on convolutional neural networks [R. Schirrmeister et al. Convolutional neural networks for EEG analysis: Design choices, training strategies, and feature visualization., *under review at Neuroimage*] were used as our reference. First, we compared the baseline learning rate schedule with different settings of the total number of epochs and initial learning rates (see Figure 4). When 30 epochs were considered, we dropped the learning rate by a factor of 10 at epoch indexes 10, 15 and 20. As expected, with more epochs used and a similar (budget proportional) schedule better results can be achieved. Alternatively, one can consider SGDR and get a similar final performance while having a better anytime performance without defining the total budget of epochs beforehand.

Similarly to our results on the CIFAR datasets, our experiments with the EEG data confirm that snapshots are useful and the median reference error (about 9%) can be improved i) by 1-2% when model snapshots of a single run are considered, and ii) by 2-3% when model snapshots from both hyperparameter settings are considered. The latter would correspond to $N = 2$ in Section (4.3).

## 4.5 PRELIMINARY EXPERIMENTS ON A DOWNSAMPLED IMAGENET DATASET

In order to additionally validate our SGDR on a larger dataset, we constructed a downsampled version of the ImageNet dataset [P. Chrabaszcz, I. Loshchilov and F. Hutter. A Downsampled Variant of ImageNet as an Alternative to the CIFAR datasets., *in preparation*]. In contrast to earlier attempts (Pouransari & Ghili, 2015), our downsampled ImageNet contains exactly the same images from 1000 classes as the original ImageNet but resized with box downsampling to $32 \times 32$ pixels. Thus, this dataset is substantially harder than the original ImageNet dataset because the average number of pixels per image is now two orders of magnitude smaller. The new dataset is also more difficult than the CIFAR datasets because more classes are used and the relevant objects to be classified often cover only a tiny subspace of the image and not most of the image as in the CIFAR datasets.

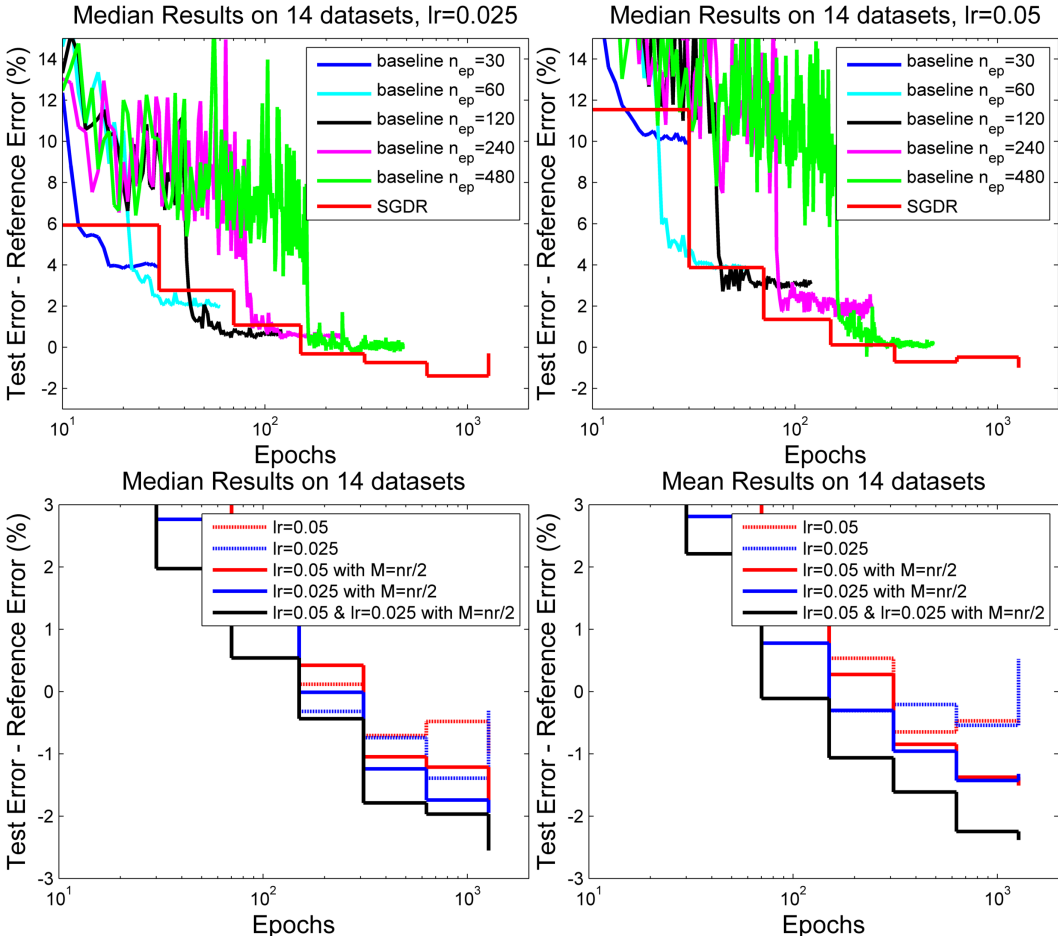

Figure 4: (**Top**) Improvements obtained by the baseline learning rate schedule and SGDR w.r.t. the best known reference classification error on a dataset of electroencephalographic (EEG) recordings of brain activity for classification of actual right and left hand and foot movements of 14 subjects with roughly 1000 trials per subject. Both considered approaches were tested with the initial learning rate $lr = 0.025$ (**Top-Left**) and $lr = 0.05$ (**Top-Right**). Note that the baseline approach is considered with different settings of the total number of epochs: 30, 60, ..., 480. (**Bottom**) SGDR with $lr = 0.025$ and $lr = 0.05$ without and with $M$ model snapshots taken at the last $M = nr/2$ restarts, where $nr$ is the total number of restarts.

We benchmarked SGD with momentum with the default learning rate schedule, SGDR with $T_0 = 1, T_{mult} = 2$ and SGDR with $T_0 = 10, T_{mult} = 2$ on WRN-28-10, all trained with 4 settings of the initial learning rate $\eta^i_{max}$: 0.050, 0.025, 0.01 and 0.005. We used the same data augmentation procedure as for the CIFAR datasets. Similarly to the results on the CIFAR datasets, Figure 5 shows that SGDR demonstrates better anytime performance. SGDR with $T_0 = 10, T_{mult} = 2, \eta^i_{max} = 0.01$ achieves top-1 error of 39.24% and top-5 error of 17.17% matching the original results by AlexNets (40.7% and 18.2%, respectively) obtained on the original ImageNet with full-size images of ca. 50 times more pixels per image (Krizhevsky et al., 2012b). Interestingly, when the dataset is permuted only within 10 subgroups each formed from 100 classes, SGDR also demonstrates better results (see Figure 8 in the Supplementary Material). An interpretation of this might be that while the initial learning rate seems to be very important, SGDR reduces the problem of improper selection of the latter by scanning / annealing from the initial learning rate to 0.

Clearly, longer runs (more than 40 epochs considered in this preliminary experiment) and hyperparameter tuning of learning rates, regularization and other hyperparameters shall further improve the results.

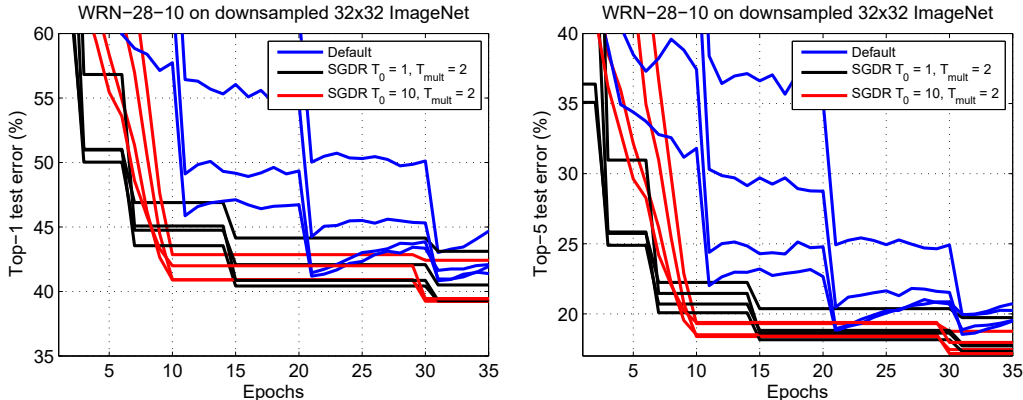

Figure 5: Top-1 and Top-5 test errors obtained by SGD with momentum with the default learning rate schedule, SGDR with $T_0 = 1, T_{mult} = 2$ and SGDR with $T_0 = 10, T_{mult} = 2$ on WRN-28-10 trained on a version of ImageNet, with all images from all 1000 classes downsampled to $32 \times 32$ pixels. The same baseline data augmentation as for the CIFAR datasets is used. Four settings of the initial learning rate are considered: 0.050, 0.025, 0.01 and 0.005.

## 5    DISCUSSION

Our results suggest that *even without any restarts* the proposed aggressive learning rate schedule given by eq. (5) is competitive w.r.t. the default schedule when training WRNs on the CIFAR-10 (e.g., for $T_0 = 200, T_{mult} = 1$) and CIFAR-100 datasets. In practice, the proposed schedule requires only two hyper-parameters to be defined: the initial learning rate and the total number of epochs.

We found that the anytime performance of SGDR remain similar when shorter epochs are considered (see section 8.1 in the Supplemenary Material).

One *should not* suppose that the parameter values used in this study and many other works with (Residual) Neural Networks are selected to demonstrate the fastest decrease of the training error. Instead, the best validation or / and test errors are in focus. Notably, the validation error is rarely used when training Residual Neural Networks because the recommendation is defined by the final solution (in our approach, the final solution of each run). One could use the validation error to determine the optimal initial learning rate and then run on the whole dataset; this could further improve results.

The main purpose of our proposed warm restart scheme for SGD is to improve its anytime performance. While we mentioned that restarts can be useful to deal with multi-modal functions, *we do not claim* that we observe any effect related to multi-modality.

As we noted earlier, one could decrease $\eta_{max}^i$ and $\eta_{min}^i$ at every new warm restart to control the amount of divergence. If new restarts are worse than the old ones w.r.t. validation error, then one might also consider going back to the last best solution and perform a new restart with adjusted hyperparameters.

Our results reproduce the finding by Huang et al. (2016a) that intermediate models generated by SGDR can be used to build efficient ensembles at no cost. This finding makes SGDR especially attractive for scenarios when ensemble building is considered.

## 6    CONCLUSION

In this paper, we investigated a simple warm restart mechanism for SGD to accelerate the training of DNNs. Our SGDR simulates warm restarts by scheduling the learning rate to achieve competitive results on CIFAR-10 and CIFAR-100 roughly two to four times faster. We also achieved new state-of-the-art results with SGDR, mainly by using even wider WRNs and ensembles of snapshots from

SGDR's trajectory. Future empirical studies should also consider the SVHN, ImageNet and MS COCO datasets, for which Residual Neural Networks showed the best results so far. Our preliminary results on a dataset of EEG recordings suggest that SGDR delivers better and better results as we carry out more restarts and use more model snapshots. The results on our downsampled ImageNet dataset suggest that SGDR might also reduce the problem of learning rate selection because the annealing and restarts of SGDR scan / consider a range of learning rate values. Future work should consider warm restarts for other popular training algorithms such as AdaDelta (Zeiler, 2012) and Adam (Kingma & Ba, 2014).

Alternative network structures should be also considered; e.g., soon after our initial arXiv report (Loshchilov & Hutter, 2016), Zhang et al. (2016); Huang et al. (2016b); Han et al. (2016) reported that WRNs models can be replaced by more memory-efficient models. Thus, it should be tested whether our results for individual models and ensembles can be further improved by using their networks instead of WRNs. Deep compression methods (Han et al., 2015) can be used to reduce the time and memory costs of DNNs and their ensembles.

## 7    ACKNOWLEDGMENTS

This work was supported by the German Research Foundation (DFG), under the BrainLinksBrain-Tools Cluster of Excellence (grant number EXC 1086). We thank Gao Huang, Kilian Quirin Weinberger, Jost Tobias Springenberg, Mark Schmidt and three anonymous reviewers for their helpful comments and suggestions.

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

# 8 SUPPLEMENTARY MATERIAL

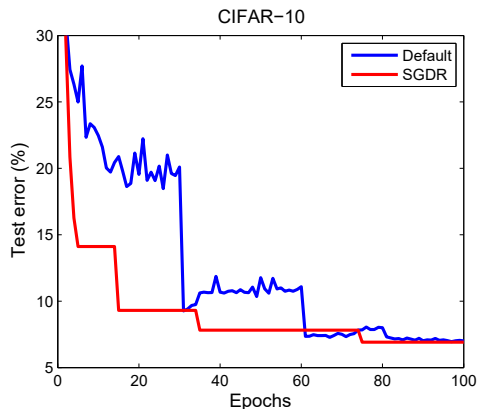

Figure 6: The median results of 5 runs for the best learning rate settings considered for WRN-28-1.

## 8.1 50K VS 100K EXAMPLES PER EPOCH

Our data augmentation procedure code is inherited from the Lasagne Recipe code for ResNets where flipped images are added to the training set. This doubles the number of training examples per epoch and thus might impact the results because hyperparameter values defined as a function of epoch index have a different meaning. While our experimental results given in Table 1 reproduced the results obtained by Zagoruyko & Komodakis (2016), here we test whether SGDR still makes sense for WRN-28-1 (i.e., ResNet with 28 layers) where one epoch corresponds to 50k training examples. We investigate different learning rate values for the default learning rate schedule (4 values out of [0.01, 0.025, 0.05, 0.1]) and SGDR (3 values out of [0.025, 0.05, 0.1]). In line with the results given in the main paper, Figure 6 suggests that SGDR is competitive in terms of anytime performance.

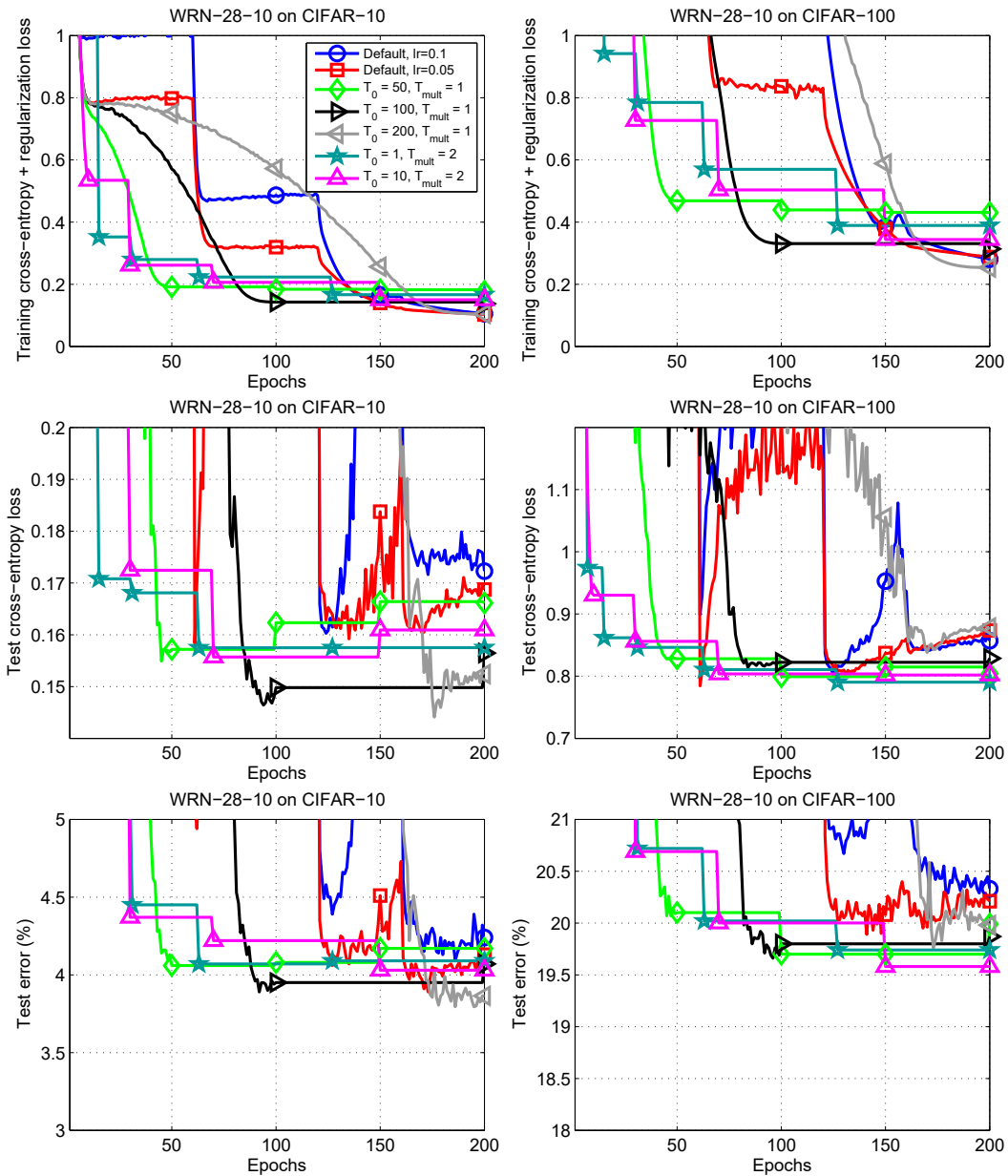

Figure 7: Training cross-entropy + regularization loss (top row), test loss (middle row) and test error (bottom row) on CIFAR-10 (left column) and CIFAR-100 (right column).

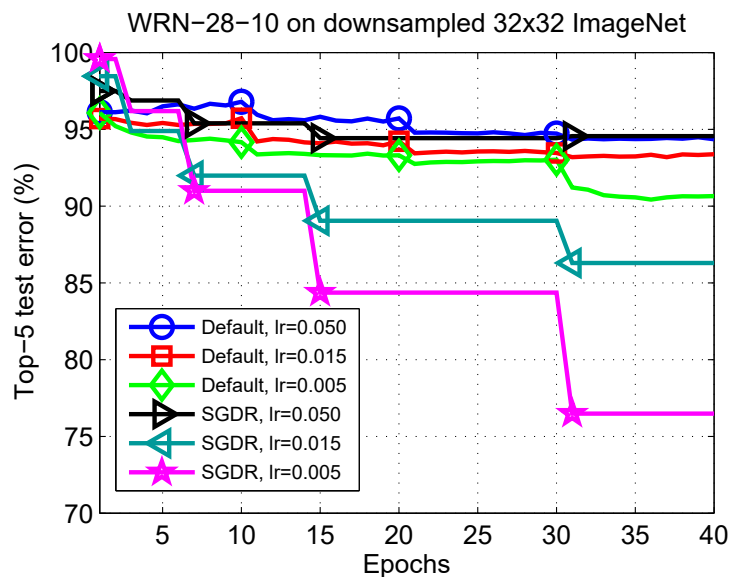

Figure 8: Top-5 test errors obtained by SGD with momentum with the default learning rate schedule and SGDR with $T_0 = 1, T_{mult} = 2$ on WRN-28-10 trained on a version of ImageNet, with all images from all 1000 classes downsampled to $32 \times 32$ pixels. The same baseline data augmentation as for the CIFAR datasets is used. Three settings of the initial learning rate are considered: 0.050, 0.015 and 0.005. In contrast to the experiments described in the main paper, here, the dataset is permuted only within 10 subgroups each formed from 100 classes which makes good generalization much harder to achieve for both algorithms. An interpretation of SGDR results given here might be that while the initial learning rate seems to be very important, SGDR reduces the problem of improper selection of the latter by scanning / annealing from the initial learning rate to 0.

