# Peer review of "SGDR: Stochastic Gradient Descent with Warm Restarts"

_ICLR 2017 — accepted_

[Official Review · AnonReviewer3 · rating 7 · confidence 4 · 15 Dec 2016]
**No Title**

This an interesting investigation into learning rate schedules, bringing in the idea of restarts, often overlooked in deep learning. The paper does a thorough study on non-trivial datasets, and while the outcomes are not fully conclusive, the results are very good and the approach is novel enough to warrant publication. 

I thank the authors for revising the paper based on my concerns.

Typos:
- “flesh” -> “flush”

[Official Review · AnonReviewer1 · rating 7 · confidence 5 · 15 Dec 2016]
**An effective method to improve convergence of neural network training**

This paper describes a way to speed up convergence through sudden increases of otherwise monotonically decreasing learning rates. Several techniques are presented in a clear way and parameterized method is proposed and evaluated on the CIFAR task. The concept is easy to understand and the authors chose state-of-the-art models to show the performance of their algorithm. The relevance of these results goes beyond image classification.


Pros:

- Simple and effective method to improve convergence
- Good evaluation on well known database


Cons:

- Connection of introduction and topic of the paper is a bit unclear
- Fig 2, 4 and 5 are hard to read. Lines are out of bounds and maybe only the best setting for T_0 and T_mult would be clearer. The baseline also doesn't seem to converge

Remarks:
An loss surface for T_0 against T_mult would be very helpful. Also understanding the relationship of network depth and the performance of this method would add value to this analysis.

[Official Review · AnonReviewer4 · rating 7 · confidence 3 · 20 Dec 2016 (modified: 20 Jan 2017)]
**Great trick worth publishing, but is there enough material for a full paper?**

This heuristic to improve gradient descent in image classification is simple and effective, but this looks to me more like a workshop track paper. Demonstration of the algorithm is limited to one task (CIFAR) and there is no theory to support it, so we do not know how it will generalize on other tasks

Working on DNNs for NLP, I find some observations in the paper opposite to my own experience. In particular, with architectures that combine a wide variety of layer types (embedding, RNN, CNN, gating), I found that ADAM-type techniques far outperform simple SGD with momentum, as they save searching for the right learning rate for each type of layer. But ADAM only works well combined with Poliak averaging, as it fluctuates a lot from one batch to another.

Revision:
-  the authors substantially improved the contents of the paper, including experiments on another set than Cifar
-  the workshop track has been modified to breakthrough work, so my recommendation for it is not longer appropriate
I have therefore improved my rating

[Author Response · Ilya Loshchilov · 14 Jan 2017]
**Rebuttal**

We thank all reviewers for their positive evaluation and their valuable comments. We've uploaded a revision to address the issues raised and briefly reply to the reviewers' concerns here.

Experiments on additional benchmarks 
==============================

In order to address the concerns of AnonReviewer4 about our method's generality, we added two more benchmarks to the main paper.

1. We had already included additional results for another domain in the appendix of the original submission. That domain is motor-control decoding based on EEG data, which, due to the large noise present in brain signals, is very different from visual object recognition in general, and CIFAR in particular. We realize that it was not the best decision to only mention those results in the supplementary material, and we have now moved them to the main paper (new Section 4.4). In fact, our work is in part motivated and funded by a project aimed at accelerating the processing of this sort of brain data (see acknowledgments).

2. We've now added a new experiment on a new downsampled version of the ImageNet dataset (new Section 4.5), which is much harder than the CIFAR benchmark because there are 1000 classes and because the target concept often only occupies a small part of the image. Due to the computational expense of training on 1 million images, these results are very preliminary -- e.g., we didn't tune hyperparameters yet. While the dataset is difficult for both the default learning rate schedule and SGDR, SGDR achieved much better results. An interpretation of this might be that, while the initial learning rate seems to be very important, SGDR reduces the problem of improper learning rate selection by quickly annealing from the initial learning rate to 0.

We would also like to point out that the simplicity and generality of SGDR let Huang et al use it to develop snapshot ensembles (

[Final Decision · Program Chairs · 06 Feb 2017]
**ICLR committee final decision**

All reviewers viewed the paper favourably, with the only criticism being that seeing how the method complements other approaches (momentum, Adam) would make the paper more complete. We encourage the authors to include such a comparison in the camera ready version of their paper.